# Fabricating strong and tough aramid fibers by small addition of carbon nanotubes

Jiajun Luo[1,2,9], Yeye Wen[1,2,9], Xiangzheng Jia [3,9], Xudong Lei[4,5,9], Zhenfei Gao[2], Muqiang Jian[2], Zhihua Xiao[1,2], Lanying Li[6], Jiangwei Zhang [7], Tao Li[2], Hongliang Dong[8], Xianqian Wu [4,5] ✉, Enlai Gao [3] ✉, Kun Jiao[1,2] ✉ & Jin Zhang [1,2] ✉

Synthetic high-performance fibers present excellent mechanical properties and promising applications in the impact protection field. However, fabricating fibers with high strength and high toughness is challenging due to their intrinsic conflicts. Herein, we report a simultaneous improvement in strength, toughness, and modulus of heterocyclic aramid fibers by 26%, 66%, and 13%, respectively, via polymerizing a small amount (0.05 wt%) of short aminated single-walled carbon nanotubes (SWNTs), achieving a tensile strength of $6.44 \pm 0.11$ GPa, a toughness of $184.0 \pm 11.4$ MJ m$^{-3}$, and a Young's modulus of $141.7 \pm 4.0$ GPa. Mechanism analyses reveal that short aminated SWNTs improve the crystallinity and orientation degree by affecting the structures of heterocyclic aramid chains around SWNTs, and in situ polymerization increases the interfacial interaction therein to promote stress transfer and suppress strain localization. These two effects account for the simultaneous improvement in strength and toughness.

Synthetic high-performance fibers, such as polyaramid, ultra-high molecular weight polyethylene (UHMWPE), and poly-p-phenylene benzobisoxazole (PBO), have largely replaced metallic protective materials due to their high mechanical performance, low density, and excellent manufacturability into textiles and composites[1-4]. The attainment of both high strength and high toughness, which are generally mutually exclusive, is vital for fiber applications[5-7]. For example, the high strength of polymer fibers is usually attributed to high crystallinity and high orientation degree of polymer chains[1,8-10]. However, these features suppress the mobility of polymer chains, which induces brittle behavior and poor toughness[11]. This trade-off between strength and toughness limits their applications in resisting high impact.

Therefore, it is challenging to develop fibers with high strength and high toughness.

Carbon nanotubes (CNTs) have long been used as reinforcements for composite materials[12-18], especially for fiber materials[19]. For modest-mechanical-performance fibrous materials, such as nylon, polypropylene, polyacrylonitrile, and polyvinyl alcohol (PVA), it has been well demonstrated that the addition of CNTs can largely improve their mechanical performance[20-26]. This is because the crystallinity and orientation degree of such materials are relatively low, leaving plenty of room for improvement. For example, the tensile strength of these fiber materials can be improved by over 50% through small addition of CNT (0.5 wt%–1.0 wt%)[22-25]. Compared to modest-mechanical-performance

[1]Beijing National Laboratory for Molecular Sciences, School of Materials Science and Engineering, College of Chemistry and Molecular Engineering, Academy for Advanced Interdisciplinary Studies, Beijing Science and Engineering Center for Nanocarbons, Peking University, 100871 Beijing, China. [2]Beijing Graphene Institute (BGI), 100095 Beijing, China. [3]Department of Engineering Mechanics, School of Civil Engineering, Wuhan University, 430072 Wuhan, China. [4]Institute of Mechanics, Chinese Academy of Sciences, 100190 Beijing, China. [5]School of Engineering Science, University of Chinese Academy of Sciences, 100049 Beijing, China. [6]China Bluestar Chengrand Chemical Co., Ltd, 611430 Chengdu, China. [7]Science Center of Energy Material and Chemistry, College of Chemistry and Chemical Engineering, Inner Mongolia University, 010021 Hohhot, China. [8]Center for High Pressure Science and Technology Advanced Research, 201203 Shanghai, China. [9]These authors contributed equally: Jiajun Luo, Yeye Wen, Xiangzheng Jia, Xudong Lei.
✉e-mail: wuxianqian@imech.ac.cn; enlaigao@whu.edu.cn; jiaokun-cnc@pku.edu.cn; jinzhang@pku.edu.cn

fibrous materials, the high crystallinity and high orientation degree of high-mechanical-performance fibers increase the challenge to achieve further optimization of structure with the addition of CNTs, since unsuitable addition of CNTs might even damage the structures of pristine high-mechanical-performance fibers[27–31].

In the pursuit of effective reinforcement by CNTs, multi-phases (e.g., CNT phase, polymer phase, interphase) and multi-scale structures (e.g., nanoscale, microscale, macroscale) of composite fibers must be considered for global optimization[32]. However, compared to global optimization, many previous studies focused on local optimization of a certain phase or scale of structures[30,31,33]. For example, it has long been believed that long CNTs are favorable for effective stress transfer, and considerable effort has been devoted to reinforcing fibers with the addition of long CNTs[15,34,35]. In fact, CNTs with a length significantly longer than the persistence length will not behave like a rigid rod, and the longer CNTs are, the more tangled up they can get, leading to challenges in dispersion and alignment[36–40]. As a result, this issue would further affect the structural integrity of polymer chains. Therefore, the dispersion, alignment, and interaction of CNTs in fiber materials, and their effects on polymer chains should be considered in balance.

Herein, we demonstrate a strategy for fabricating short aminated single-walled carbon nanotube (sa-SWNT) reinforced heterocyclic aramid fibers (HAFs) with high strength and high toughness by in situ polymerization and wet spinning. The sa-SWNTs were introduced into the polymerization system to copolymerize with heterocyclic aramid monomers. Subsequently, the spinning dopes were processed to prepare the sa-SWNT/heterocyclic aramid composite fibers (sa-SWNT-HAFs) (Fig. 1a). The sa-SWNT with a length near its persistence length shows good dispersity and rod-like behavior, which significantly improve the crystallinity and orientation degree of HAFs. The in situ polymerization between sa-SWNTs and heterocyclic aramid

monomers increases the interfacial interaction to promote stress transfer and prevent strain localization. As a result, strong (tensile strength of 6.44 ± 0.11 GPa) and tough (toughness of 184.0 ± 11.4 MJ m⁻³) HAFs are fabricated (Fig. 1b, c). Moreover, sa-SWNT-HAFs possess an ultra-high strength of 7.36 GPa at a high strain rate of 1400 s⁻¹ and superior dynamic energy absorption capacity, demonstrating their excellent dynamic mechanical responses and potential applications in the fields of impact protection and high-tenacity composites (Fig. 1d).

## Results

### Structural design and characterization of sa-SWNTs

In order to achieve the uniform dispersion, optimal alignment and strong interaction of SWNTs in HAFs, we report a four-step method to prepare sa-SWNTs (Fig. 2a). The transmission electron microscope (TEM) and scanning electron microscope (SEM) images of raw SWNTs indicate that they have a diameter of around 3 nm and an average length of around 6.72 µm (Fig. 2b, c, Supplementary Figs. 1a, 2a). These raw SWNTs prefer to be bundled because of their large aspect ratio and strong intertube interaction. Through the process of primary oxidation, long-SWNTs (average length of around 1.66 µm) decorated with oxygen-containing functional groups were obtained (Supplementary Figs. 1b, e, 2b, 3a). After violent ultrasonication of the long-SWNT dispersion, the length of SWNTs was further reduced, and the resultant short-SWNTs (average length of around 0.66 µm) exhibit a good dispersity (Supplementary Figs. 1c, 2c, 4). The reoxidation of short-SWNTs was performed to prepare short carboxyl SWNTs (sc-SWNTs, average length of around 0.63 µm) with more reactive carboxyl groups (Supplementary Figs. 1d, 2d, 3, 4). In order to construct strong covalent bonding between SWNTs and heterocyclic aramid chains, amino groups which can react with heterocyclic aramid monomers were introduced to SWNTs. Thus, sa-SWNTs were prepared by carrying out an amination reaction of sc-SWNTs with ethylenediamine (Supplementary Fig. 3e).

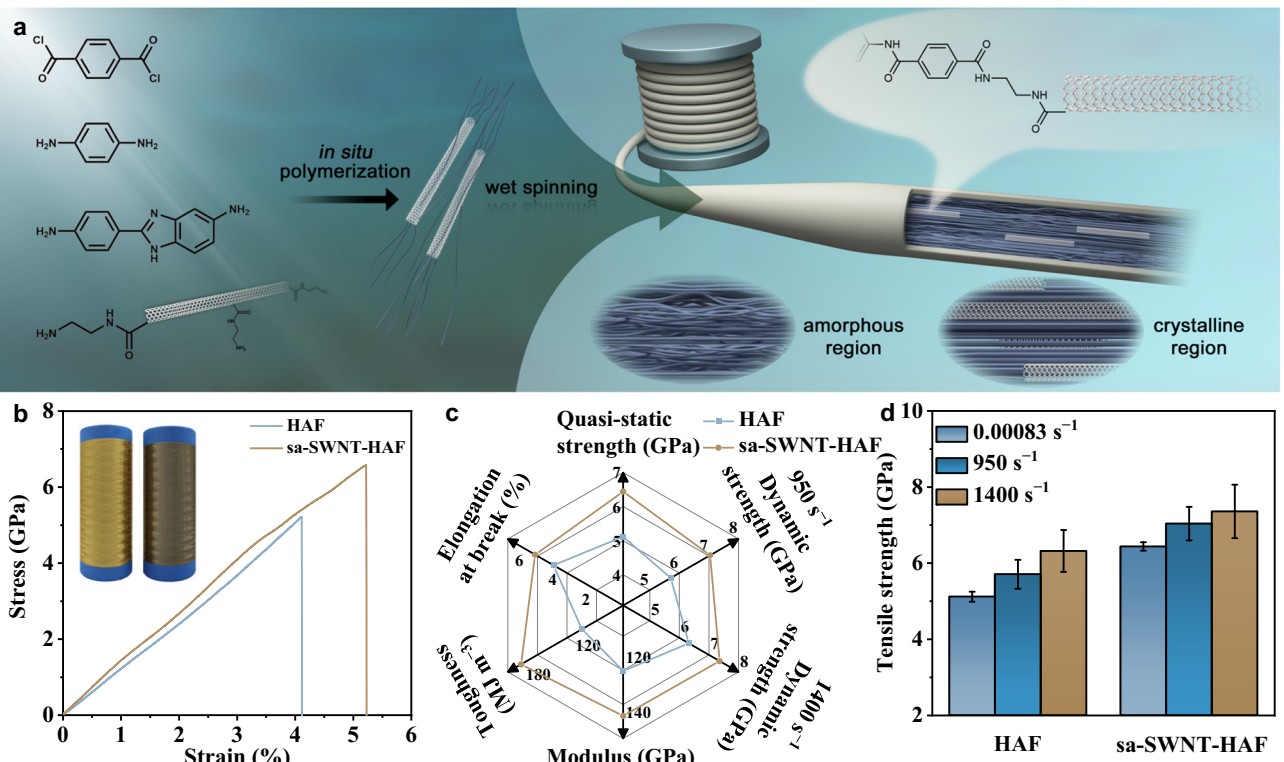

**Fig. 1 | Preparation and properties of composite fibers. a** Schematic diagram of the polymerization and spinning process of composite fibers. **b** Stress-strain curves of HAFs and sa-SWNT-HAFs. The inset shows the digital photograph of HAFs and sa-SWNT-HAFs. **c** A radial plot comparing the mechanical properties of HAFs and sa-SWNT-HAFs. **d** Comparison of the tensile strength of HAFs and sa-SWNT-HAFs at different strain rates. Error bars indicate the standard deviation of the tensile strength.

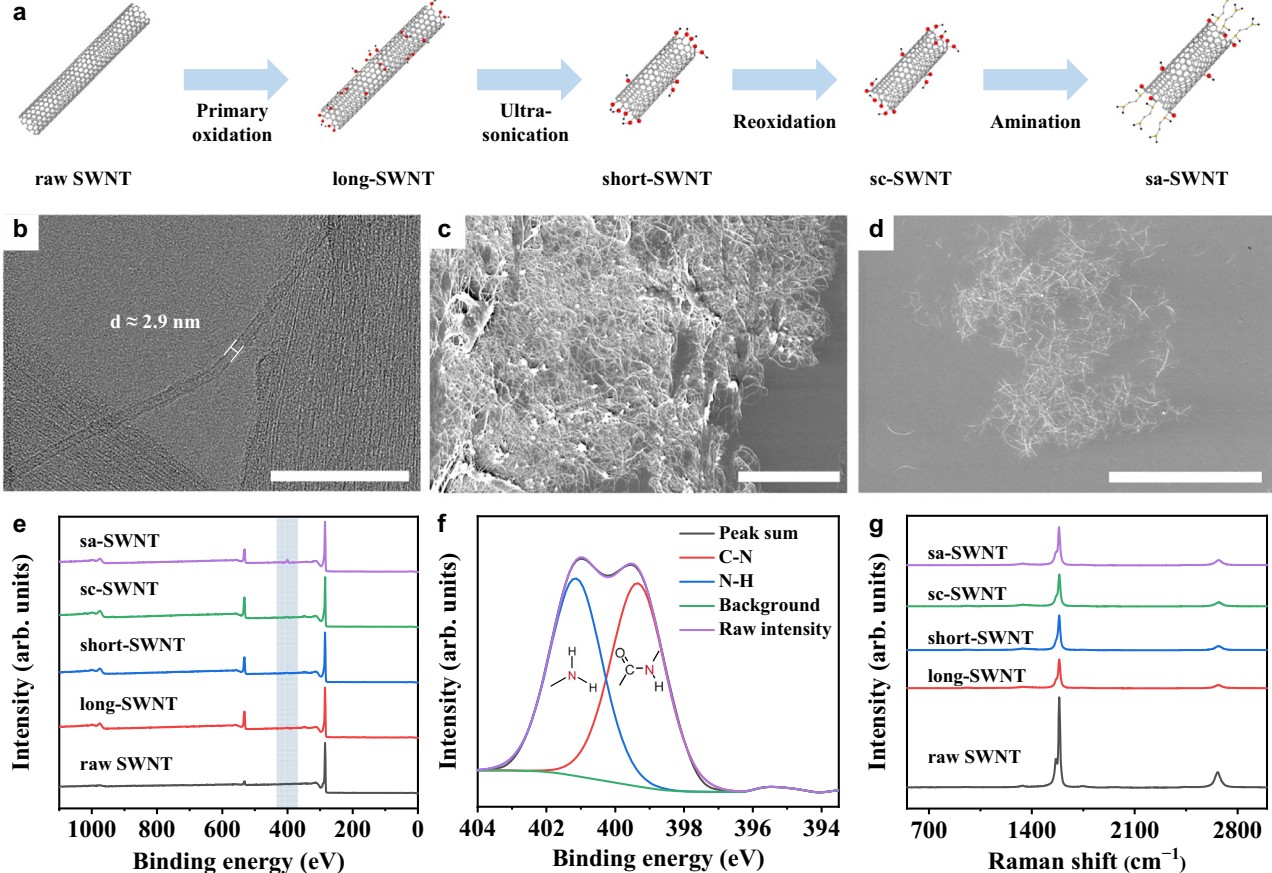

**Fig. 2 | Structural design and characterization of SWNTs. a** Schematic diagram of the fabrication steps of sa-SWNTs. **b** High-resolution transmission electron microscope (TEM) image of raw SWNTs. Scale bar, 50 nm. **c** Scanning electron microscope (SEM) image of raw SWNTs. Scale bar, 10 μm. **d** SEM image of sa-SWNTs. Scale bar, 10 μm. **e** X-ray photoelectron spectroscopy (XPS) spectra of different SWNTs. The blue stripe shows the position of N $1s$ peak. **f** XPS spectrum of N $1s$ of sa-SWNTs. **g** Raman spectra of different SWNTs.

The results indicate that sa-SWNTs show a good dispersity, their lengths are mostly distributed between 0.4 μm and 1.0 μm (average length of around 0.62 μm), which are near the persistence length (Fig. 2d, Supplementary Figs. 1, 2e, 4 and Supplementary Note 1)[36].

Importantly, similar contents of the nitrogen element in amide groups and amino groups of sa-SWNTs indicate that each ethylenediamine molecule tends to react with one carboxyl group of sc-SWNT and generates a free reactive amino group, rather than crosslinking two sc-SWNTs (Fig. 2e, f, Supplementary Note 2). After the two-step oxidation, the ratio of the intensities of the $G$ and $D$ peaks ($G/D$; indicating intact structure and defects in the CNTs, respectively) calculated from Raman spectra of treated SWNTs is reduced, but still is as high as about 15, which indicates that their tube walls remain relatively intact (Fig. 2g, Supplementary Fig. 5)[41]. Therefore, it is reasonable to speculate that most carboxyl groups as well as the amino groups were decorated on the port of SWNTs.

## Fabrication of sa-SWNT-HAFs through wet spinning

As a special monomer, the as-prepared sa-SWNTs were added into the polymerization system containing heterocyclic aramid monomers [*p*-phenylenediamine, 2-(4-Aminophenyl)-1H-benzimidazol-5-amine and terephthaloyl chloride (TPC)] to prepare spinning dopes (Supplementary Figs. 6a, 7). According to the molecular weight tests of heterocyclic aramid chains by gel permeation chromatography (GPC) method, the addition of SWNTs does not affect the polymerization of heterocyclic aramid monomers (Supplementary Fig. 8, Supplementary Table 1). To verify that sa-SWNTs can bridge heterocyclic aramid chains by forming covalent bonding, TPC monomer was selected to

react with sa-SWNTs. X-ray photoelectron spectroscopy (XPS) high-resolution N $1s$ spectrum of the resultant sa-SWNTs shows an almost complete conversion of free amino groups to amide bonds, indicating that sa-SWNTs can participate in the copolymerization reaction as special monomers (Fig. 1a, Supplementary Fig. 9).

Subsequently, all fibers were prepared by the same optimal process parameters as HAFs through a wet-spinning system (Supplementary Figs. 6b, 10). The dispersant (polyvinylpyrrolidone) can be removed during the spinning process (Supplementary Fig. 11, Supplementary Note 3). For clarity, the HAFs composited with aminated long-SWNTs (al-SWNTs), aminated short-SWNTs (as-SWNTs), and sc-SWNTs are abbreviated as al-SWNT-HAFs, as-SWNT-HAFs, and sc-SWNT-HAFs, respectively. Particularly, after 6 h of wet spinning process, there is no SWNT aggregation from the spinning dopes of sc-SWNT/heterocyclic aramid and sa-SWNT/heterocyclic aramid on the thin filter mesh, indicating the uniform dispersion of sc-SWNTs and sa-SWNTs in spinning dopes (Supplementary Fig. 12).

## Structural characterization of sa-SWNT-HAFs

Due to the same spinning process, the surface morphologies of HAFs and sa-SWNT-HAFs are nearly consistent, which further proves that the addition of sa-SWNTs does not affect the preparation of fibers (Fig. 3a, b). The TEM image of the axial cross-section of sa-SWNT-HAFs shows that sa-SWNTs well align along the axial direction of HAFs (Fig. 3c). This result is attributed to the rod-like behavior of sa-SWNTs with a length near the persistence length. Due to the long length and poor dispersity of al-SWNTs, they bend inside al-SWNT-HAFs and aggregate to form bundles (Supplementary Fig. 13). The three-dimensional (3D) void

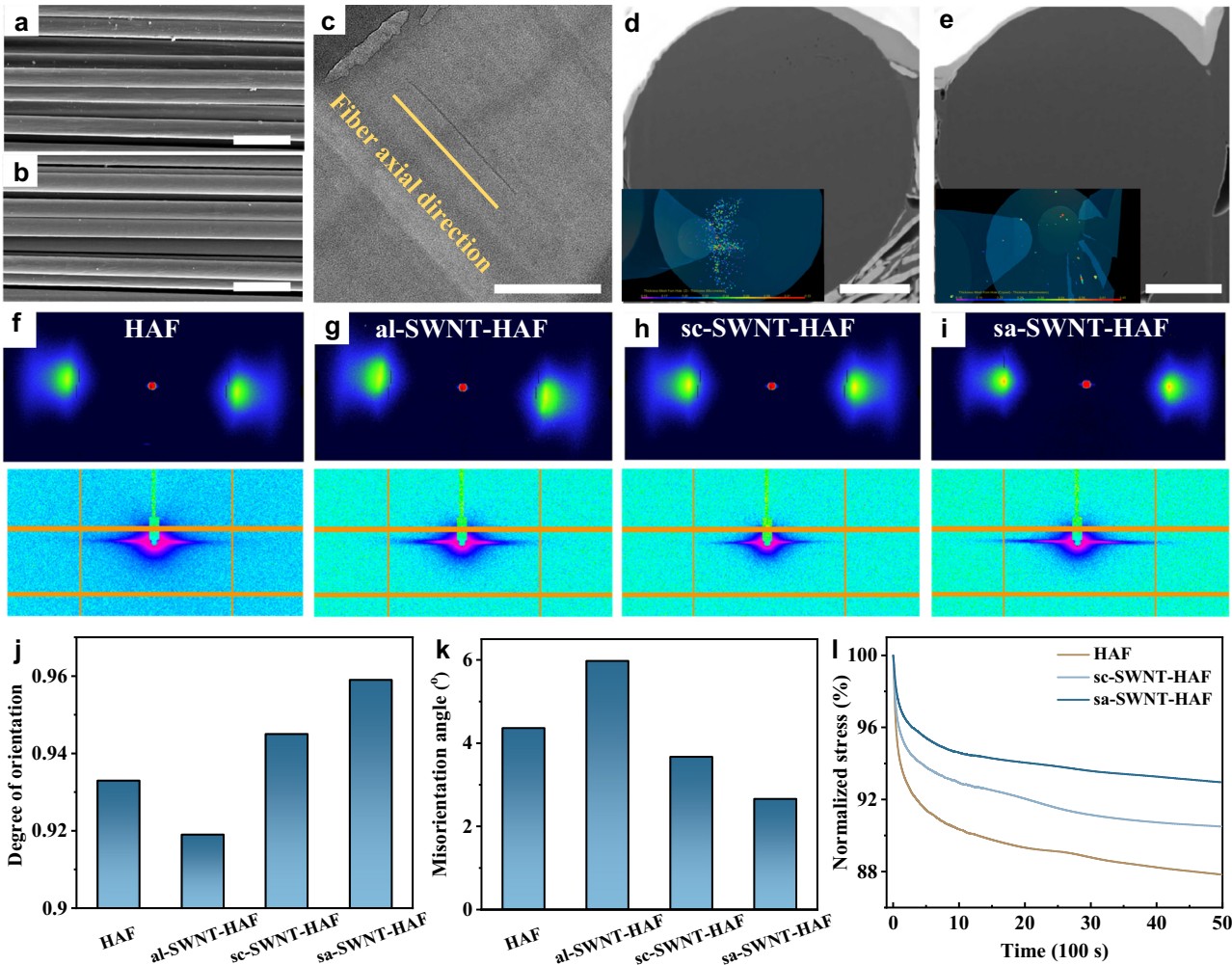

**Fig. 3 | Structural characterization of composite fibers.** SEM images of (**a**) HAFs and (**b**) sa-SWNT-HAFs. **c** TEM image of the axial cross-section of sa-SWNT-HAFs. SEM images of the radial cross-section of (**d**) HAFs and (**e**) sa-SWNT-HAFs. The inset shows the 3D-reconstructed void microstructure derived from focused ion beam and SEM tomography (FIB-SEMT). Scale bars, 50 μm in (**a**, **b**); 200 nm in (**c**); 5 μm in (**d**, **e**). 2D wide angle X-ray scattering (2D-WAXS) patterns (up) and 2D small angle X-ray scattering (2D-SAXS) patterns (down) of (**f**) HAFs, (**g**) al-SWNT-HAFs, (**h**) sc-SWNT-HAFs, and (**i**) sa-SWNT-HAFs. **j** Comparison of the orientation degree of different fibers derived from 2D-WAXS analysis. **k** Comparison of the microfiber misorientation angle of different fibers derived from 2D-SAXS analysis. **l** Stress relaxation curves of different fibers at 1.5% strain.

microstructures of HAFs and sa-SWNT-HAFs were reconstructed using focused ion beam (FIB) and SEM tomography (FIB-SEMT; Fig. 3d, e, Supplementary Movies 1–4). The results show that sa-SWNT-HAFs have a lower porosity in comparison with HAFs (Supplementary Fig. 14, Supplementary Note 4).

Wide angle X-ray scattering (WAXS) experiments were executed on the fibers to evaluate the crystallinity and orientation degree of materials (Fig. 3f–i). The results demonstrate that the addition of sa-SWNTs contributes to a significant decrease in the full width at half maxima (FWHM) of radial integration curves in the equatorial of the WAXS patterns (from 2.32 for HAFs to 1.84 for sa-SWNT-HAFs), indicating that the crystallinity of sa-SWNT-HAFs is greatly improved compared with HAFs (Supplementary Fig. 15a, b). By integrating the curve against the azimuthal degree of WAXS patterns, the calculated crystalline orientation degree of sa-SWNT-HAFs (0.959) is higher than that of HAFs (0.933), indicating that the orientation degree of sa-SWNT-HAFs is improved by sa-SWNTs (Fig. 3j, Supplementary Fig. 15c). Due to the inability to form covalent bonds with heterocyclic aramid chains, sc-SWNTs have a weak effect on the structure of heterocyclic aramid chains. Thus, the crystallinity and orientation degree of sc-SWNT-HAFs are less than those of sa-SWNT-HAFs. In addition, the misorientation angle of microfibers derived from the patterns of small

angle X-ray scattering (SAXS) can be used to probe the orientation of the microstructure in fibers (Fig. 3f–i). The sa-SWNT-HAFs exhibit a low derived misorientation angle (2.66°) of microfibers compared with HAFs (4.36°), al-SWNT-HAFs (5.98°), and sc-SWNT-HAFs (3.67°), which indicates that the arrangement of microfibers in sa-SWNT-HAFs is more ordered (Fig. 3k). It is worth noting that the crystallization and orientation of heterocyclic aramid chains can be inhibited by the aggregation and bending structure of al-SWNTs in al-SWNT-HAFs (Fig. 3g, j, k). Moreover, the crystallinity and orientation degree of as-SWNT-HAFs decrease significantly compared with sa-SWNT-HAFs (Supplementary Fig. 16). Therefore, the reoxidation procedure of short-SWNTs is necessary to increase the number of functional groups and improve their dispersity for constructing favorable composite structures. As a result, the short and aminated structural design of sa-SWNTs is essential for increasing the crystallinity and orientation degree of sa-SWNT-HAFs, which also accounts for their low porosity. Importantly, too much addition of sa-SWNTs has an adverse impact on the structure of heterocyclic aramid chains and could increase the misorientation of microfibers, which might be originated from the aggregation of sa-SWNTs (Supplementary Figs. 17, 18). Due to the advantageous structures of sa-SWNTs, the small addition of them can realize the significant structural optimization of composite fibers.

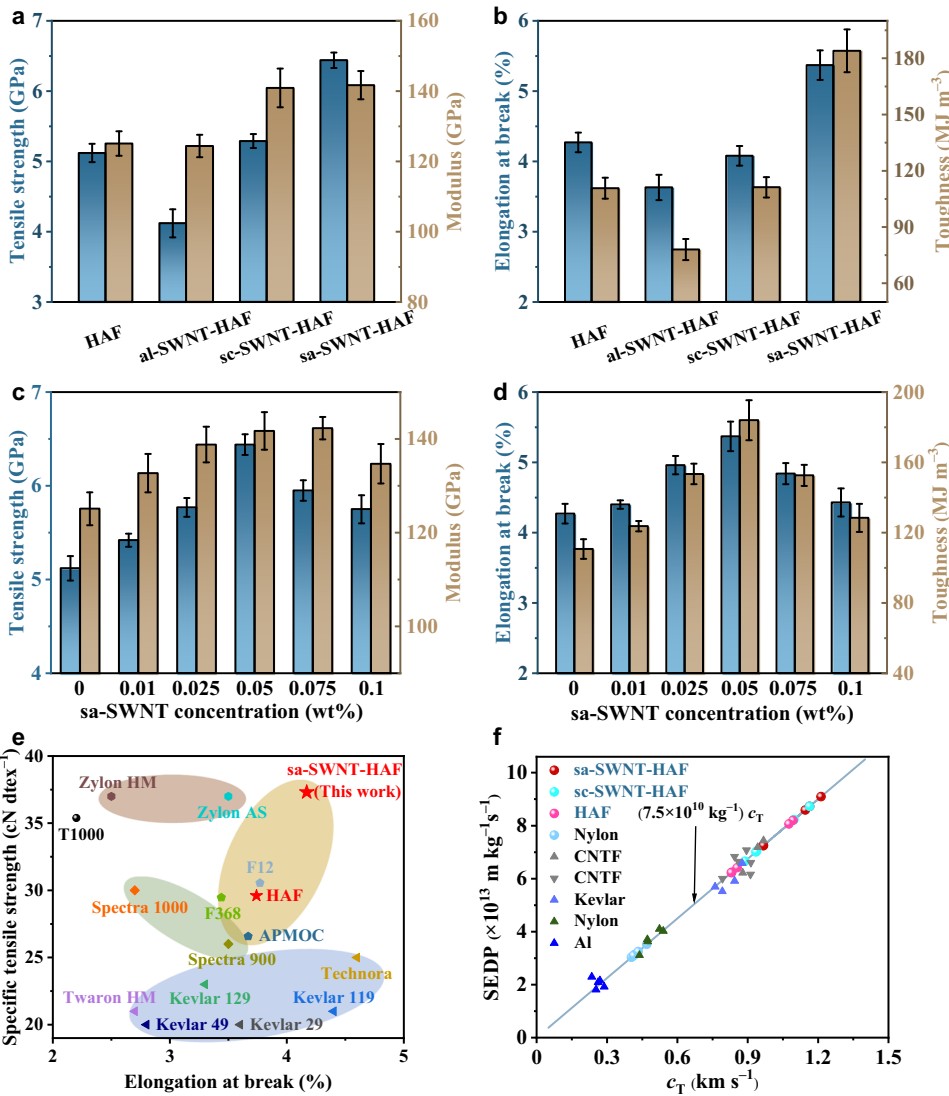

**Fig. 4 | Mechanical properties of composite fibers. a** Comparison of tensile strength and modulus of different fibers. **b** Comparison of elongation at break and toughness of different fibers. **c** Comparison of tensile strength and modulus of sa-SWNT-HAFs with different concentrations of sa-SWNTs. **d** Comparison of elongation at break and toughness of sa-SWNT-HAFs with different concentrations of sa-SWNTs. All error bars indicate the standard deviation. **e** Comparison of specific tensile strength and elongation at break between commercial high-performance fibers and our prepared fibers. The measured data were collected from the test of multifilament. **f** Specific energy dissipation power (SEDP) of different fibers. The data of triangle symbols in the figure were collected from literature[46].

Stress relaxation experiments and fiber dissolution experiments were conducted to characterize the interfacial strength of fibers. Stress relaxation curves show that the retention of the initial stress of sa-SWNT-HAFs (93.0%) is higher than those of sc-SWNT-HAFs (90.5%) and HAFs (87.8%) (Fig. 3l). The higher resistance to the interchain slippage of sa-SWNT-HAFs is originated from their strong interfacial interaction based on covalent bonding between sa-SWNTs and heterocyclic aramid chains and the compact structure. Moreover, sa-SWNT-HAFs have higher resistance to dissolution into nanofibers in a potassium hydroxide (KOH)/dimethyl sulfoxide system in comparison with sc-SWNT-HAFs and HAFs, implying a stronger interaction between nanofibers and the compact structure of sa-SWNT-HAFs (Supplementary Fig. 19).

## Mechanical performances of sa-SWNT-HAFs

The mechanical properties of sa-SWNT-HAFs were probed by testing monofilament and the results show that the optimal amount of sa-SWNTs addition (0.05 wt%) can prepare high-performance sa-SWNT-HAFs with a tensile strength of 6.44 ± 0.11 GPa, a modulus of 141.7 ± 4.0 GPa, an elongation at break of 5.37 ± 0.21%, and a toughness of 184.0 ± 11.4 MJ m$^{-3}$, displaying increases of 26%, 13%, 26%, and 66% compared with HAFs, respectively (Fig. 4a, b, Supplementary Fig. 20a, e and Supplementary Tables 2, 3). The sa-SWNT-HAFs with other concentrations of sa-SWNTs show inferior performances due to low crystallinity and orientation degree resulting from an inadequate amount or aggregation of sa-SWNTs (Fig. 4c, d, Supplementary Fig. 21 and Supplementary Tables 4–7). For comparison, the tensile strength of al-SWNT-HAFs with low crystallinity and orientation degree is 4.12 ± 0.20 GPa and the toughness is 78.0 ± 5.6 MJ m$^{-3}$ (Supplementary Fig. 20b, Supplementary Table 8). The sc-SWNT-HAFs without strong interfacial interaction have a tensile strength of 5.29 ± 0.10 GPa and a toughness of 111.2 ± 5.4 MJ m$^{-3}$ (Supplementary Fig. 20d, Supplementary Table 9). Due to the limited amino content and poor dispersity of as-SWNTs, the as-SWNT-HAFs exhibit a tensile strength of 5.85 ± 0.14 GPa and a toughness of 138.3 ± 10.3 MJ m$^{-3}$ (Supplementary Fig. 20c, Supplementary Table 10).

We summarized the reported tensile strength and elongation at break of various high-performance fibers reinforced by CNTs

(Supplementary Fig. 22, Supplementary Table 11). Due to the global optimization strategy in our work, which considers the structure of CNTs, their interaction and effect on the polymer chains in balance, the small addition of sa-SWNTs can achieve the most effective reinforcement in comparison with other reported work. Moreover, the tensile strength and toughness of HAFs and sa-SWNT-HAFs were compared with those reported in literature (Supplementary Fig. 23, Supplementary Table 12). The results showed that sa-SWNT-HAFs prepared in our work exhibit better mechanical performances[42].

Compared to the mechanical properties of monofilament, the mechanical properties of multifilament are used to evaluate the application potential of fibers. Hence, the mechanical properties of yarns were also measured (Supplementary Figs. 24–26, Supplementary Tables 13, 14). The results indicate that the optimal sa-SWNT-HAF yarns have a specific tensile strength of $37.31 \pm 1.07$ cN dtex$^{-1}$, a modulus of $925.64 \pm 15.97$ cN dtex$^{-1}$, and an elongation at break of $4.17 \pm 0.07\%$, displaying increases of 26%, 9%, and 12% compared with HAF yarns, respectively (Supplementary Figs. 24, 26, Supplementary Table 13). In comparison, the mechanical properties of al-SWNT-HAF yarns, as-SWNT-HAF yarns, and sc-SWNT-HAF yarns were also measured (Supplementary Figs. 24, 26, Supplementary Table 14). These results are consistent with those of monofilaments, which indicates the uniformity of our prepared fibers. Furthermore, the mechanical properties of our HAF yarns and sa-SWNT-HAF yarns were compared with commercial high-performance fiber yarns, such as HAFs, p-aramid fibers, carbon fibers, PBO fibers, UHMWPE fibers[43–45] (Fig. 4e, Supplementary Fig. 27 and Supplementary Table 15). The results indicate that the performance of our pristine HAFs is consistent with commercial HAFs, and sa-SWNT-HAFs exhibit superior specific tensile strength and superior elongation at break. In addition, our fibers not only have a higher elongation at break, but also show a higher modulus than most aramid fibers. Therefore, sa-SWNT-HAFs hold great promise for applications in impact protection and impact-resistant composites.

To understand the mechanical behavior of fibers under various loading rates, small-scale tensile testing was performed (Supplementary Fig. 28). The results show that these fibers exhibit a distinct strain-rate strengthening effect (Supplementary Fig. 29). With the increase of strain rate, the strength of these fibers becomes higher. Especially, the strength of sa-SWNT-HAFs is as high as 7.36 GPa at a strain rate of 1400 s$^{-1}$, which is much higher than those of HAFs and sc-SWNT-HAFs. The fibrillation and the ultimate fracture of these fibers are the main failure modes during tension. In comparison with HAFs and sc-SWNT-HAFs, the fibrillation behavior of sa-SWNT-HAFs is more obvious in realizing high strength under a high strain rate as evidenced by the more branched failure morphology (Supplementary Fig. 30). The results imply that the highest strength of sa-SWNT-HAFs among these fibers is derived from strong interfacial interactions between heterocyclic aramid chains and sa-SWNTs, and high interfacial strength between highly ordered in-line nanofibers, as evidenced by the less damaged sheath-core layer of sa-SWNT-HAFs (Supplementary Fig. 31). The strength of sa-SWNT-HAFs could be even higher under ballistic impact due to their higher local loading rates, making the sa-SWNT-HAFs a promising bulletproof material.

To investigate the dynamic energy absorption capacity of these fibers, which is used to evaluate the performance of fibers under practical impact situations, the specific energy dissipation power (SEDP) was measured by a laser-induced impact testing (Supplementary Fig. 32). SEDP is the maximum slope of $E^*$, which is calculated as

$$E^*(t) \equiv \frac{E_k(0) - E_k(t)}{E_k(0)\lambda} \qquad (1)$$

where $\lambda$, $E_k(0)$, and $E_k(t)$ are the linear mass density of the fibers, the initial kinetic energy, and the kinetic energy at time $t$, respectively. The SEDP is positively correlated with $c_T$ for impact velocity near

500 m s$^{-1}$,[46], where $c_T$ is the Euler transverse wave speed. In the present study, the impact velocity ranges from 400 to 500 m s$^{-1}$ by changing the laser energy, and the $c_T$ is estimated by a high-speed imaging system (Supplementary Fig. 33). The measured SEDP values of the nylon fiber agree with the previous study[46], validating the current experimental method. The average SEDP values of HAFs, sc-SWNT-HAFs, and sa-SWNT-HAFs are $7.03 \times 10^{13}$ m kg$^{-1}$ s$^{-1}$, $7.53 \times 10^{13}$ m kg$^{-1}$ s$^{-1}$, and $8.18 \times 10^{13}$ m kg$^{-1}$ s$^{-1}$, respectively, indicating that sa-SWNT-HAFs have the highest impact resistance (Fig. 4f, Supplementary Fig. 34). Moreover, the SEPD of sa-SWNT-HAFs is much higher than those of Al, Kevlar, and CNT fibers[46] due to the high longitudinal wave velocity $c_L = (E/\rho)^{1/2}$, where $E$ and $\rho$ are the Young's modulus and the density of the single fiber. Larger longitudinal wave velocities provide higher transverse wave speeds at a given impact velocity, allowing the fast spread of local impulse loading during impact[47]. Therefore, sa-SWNT-HAFs with high strength and high toughness have great potential in bulletproof fields.

## Mechanism discussion on the high mechanical performances of sa-SWNT-HAFs

Raman spectroscopy is commonly used to understand the enhancement mechanism of carbon-based composites[12,48]. However, due to the low concentration of SWNTs (0.05 wt%) and high fluorescence intensity of HAF in our work, it is difficult to detect the Raman signals of SWNTs in our composite fibers. To understand the enhanced mechanical performance of sa-SWNT-HAFs, we first did analyses for the modulus enhancement of our composite fibers. Classical rule-of-mixture is widely used to predict the modulus of composites based on the weighted contributions from the filler and the matrix[49]. However, this theory only predicts a 0.2% increase in Young's modulus by small addition (0.05 wt%) of sa-SWNTs. In fact, the HAFs can be divided into crystalline and amorphous regions, and the above experiments have demonstrated that sa-SWNTs can act as nucleating agents for polymer crystallization and templates for polymer orientation. Hence, we propose a modified rule-of-mixture to account for the modulus enhancement, which predicts a wide influence range (12.2 nm, Fig. 5a) of the amorphous region around sa-SWNTs into the crystalline region (see Methods for details). To support this prediction, we performed atomistic simulations. Without loss of generality, a model of polymer chains onto an sp$^2$ carbon sheet was adopted to reduce computational costs. After equilibration at room temperature (298 K) for 2 ns, the heterocyclic aramid chains around the sp$^2$ carbon sheet exhibit crystalline features. The mass density profile indicates that heterocyclic aramid chains around SWNTs within a distance of 8.6 nm are affected (Fig. 5b, Supplementary Fig. 35), which can be attributed to the high binding energy of heterocyclic aramid chains onto the sp$^2$ carbon sheet as compared with other typical polymer chains (Fig. 5c). Hence, these results from the prediction and atomistic simulations are generally consistent, indicating that large-scale orientation and crystallization of heterocyclic aramid chains by small addition of sa-SWNTs can explain the significant increase in Young's modulus of sa-SWNT-HAFs.

Next, we explored the mechanism for the increase in elongation at break of our composite fibers by performing tensile simulations. Upon stretching, HAFs and sc-SWNT-HAFs are first elastically deformed, and then a crack nucleates and propagates from the weakest link, which causes strain localizations and failure. This occurs due to that the poor shear strength between heterocyclic aramid chains, and between heterocyclic aramid chains and SWNTs can't suppress the crack propagation. Compared with pristine HAFs and sc-SWNT-HAFs, sa-SWNT-HAFs can suppress the crack nucleation/growth, which induces an increase in elongation at break (Fig. 5d). This is because in situ polymerization between heterocyclic aramid chains and sa-SWNTs largely increases the interfacial interaction. Such toughening mechanism has also been observed in previous CNT/graphene nanocomposites[50,51]. To summarize, we have uncovered the mechanism for the increase in

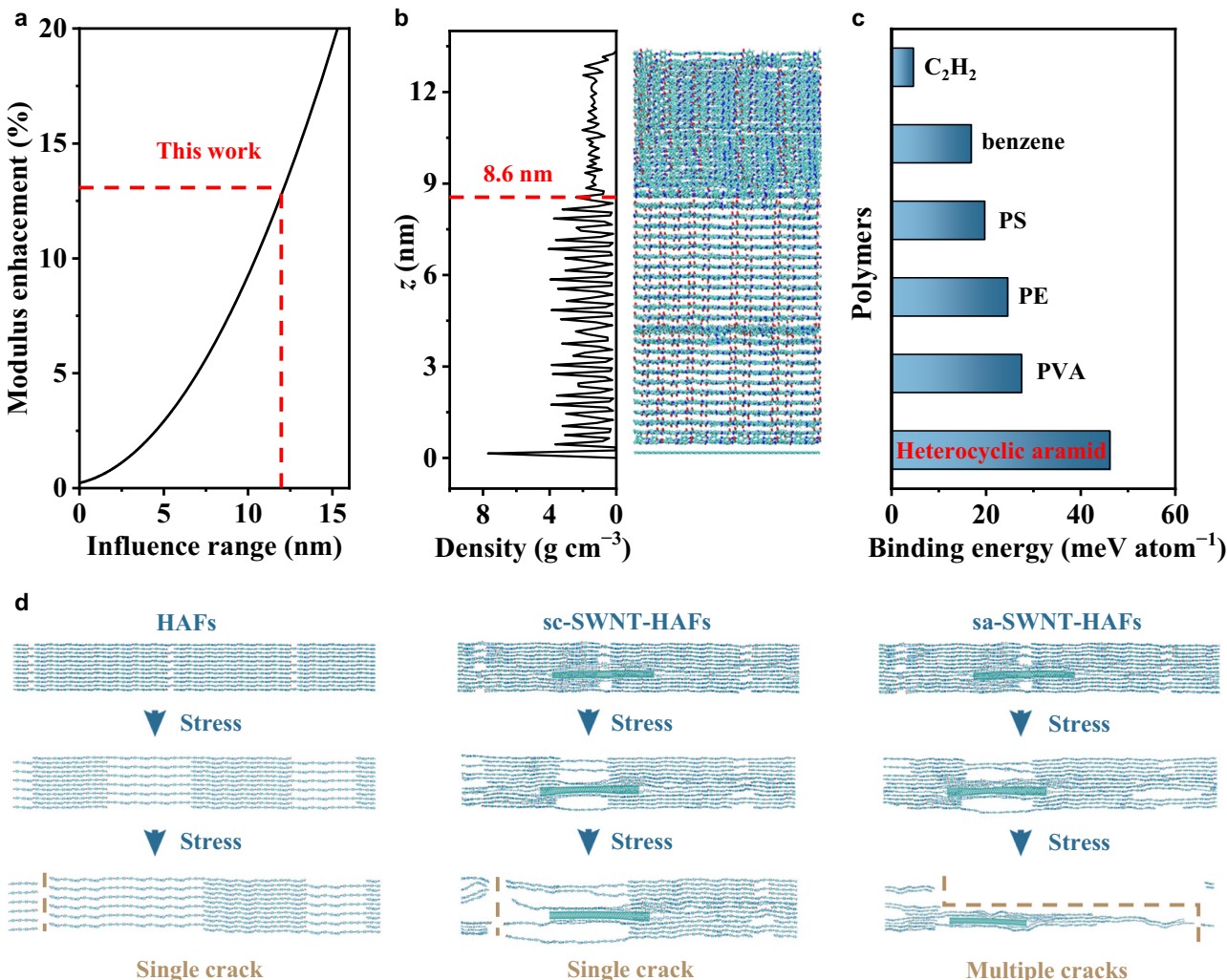

**Fig. 5 | Enhancement mechanism of sa-SWNT-HAF. a** Relation between influence range of crystallization region and the increase in Young's modulus. **b** Mass density profile and snapshots of heterocyclic aramid chains affected by an $sp^2$ carbon sheet at room temperature. **c** Binding energies of typical polymer chains onto an $sp^2$ carbon sheet. **d** Simulation snapshots of HAFs, sc-SWNT-HAFs, and sa-SWNT-HAFs under tension.

Young's modulus and elongation at break, which jointly explain the simultaneously enhanced strength and toughness of sa-SWNT-HAFs (Fig. 1b, c).

## Discussion

In summary, we prepare strong (tensile strength of 6.44 ± 0.11 GPa) and tough (toughness of 184.0 ± 11.4 MJ m$^{-3}$) HAFs through a global optimization strategy. By in situ polymerizing small amount (0.05 wt%) of sa-SWNTs into HAFs, the strength and toughness of pristine high-performance HAFs are remarkably improved by 26% and 66%, respectively. Mechanism analyses demonstrate that sa-SWNTs with good dispersity and alignment significantly improve the crystallinity and orientation degree for a large range of heterocyclic aramid chains, and the in situ polymerization between sa-SWNTs and heterocyclic aramid monomers promotes stress transfer and suppresses strain localization, which account for the simultaneous improvement in strength and toughness of sa-SWNT-HAFs. The superior mechanical performances of sa-SWNT-HAFs demonstrate their promising applications in the fields of impact protection and high-tenacity composites.

## Methods
### Materials
All chemicals, including dimethylacetamide (DMAc, ≥99%) with water content lower than 50 ppm, single-walled carbon nanotube (SWNT,

>95%), lithium chloride (LiCl, >99%), ethylenediamine (>99%), polyvinylpyrrolidone (K30), nitric acid (HNO$_3$, 65–68%), dimethyl sulfoxide (DMSO, >99%), terephthaloyl chloride (TPC, >99%), $p$-phenylenediamine (PPD, >99%) and 2-(4-Aminophenyl)-1H-benzimidazol-5-amine (PABZ, ≥99.5%) were purchased from suppliers. The raw SWNTs were purchased from Jiangsu XFNANO Materials Tech Co., Ltd. All reagents were used as received without further purification.

### Preparation of long-SWNT dispersion, short-SWNT dispersion and sc-SWNT dispersion
SWNTs (0.5 g) were dispersed in concentrated HNO$_3$ (65% w/w, 250 mL) by sonication for 1 h. After sonication, the SWNT-HNO$_3$ mixture was refluxed at 140 °C for 4 h under vigorous stirring. The as-prepared solution was then washed 4–5 times using deionized water to remove the residual HNO$_3$ and impurities by centrifugation at 10,000 $g$ for 15 min. Then, the resultant long-SWNT was dispersed in deionized water and purified by dialysis to remove the residual HNO$_3$. Subsequently, the short-SWNT was prepared by violently sonicating the long-SWNT dispersion for 3 h in an ice-water bath using an Ultrasonic Cell Disruption System with 60% power (1500 W). Next, the short-SWNT was reoxidized in HNO$_3$ solution (40% w/w, 250 mL) at 90 °C for 2 h while stirring vigorously and refluxing. Subsequently, the as-prepared sc-SWNT dispersion was washed by centrifuging (10,000 $g$ for 15 min) and dialysis to remove the residual HNO$_3$.

## Preparation of sa-SWNT dispersion

Excess ethylenediamine was added into sc-SWNT dispersion to react with the carboxyl groups of sc-SWNTs. The amination process was conducted at 90 °C for 12 h while stirring vigorously and refluxing. Next, the as-prepared solution was purified through dialysis to remove the residual ethylenediamine. The al-SWNT dispersion was prepared through the animation of long-SWNT under the same procedures. The as-SWNT solution was prepared through the animation of short-SWNT under the same procedures.

## In situ synthesis of sa-SWNT/heterocyclic aramid spinning dope

The sa-SWNT dispersion was mixed with polyvinylpyrrolidone ($m_{\text{sa-SWNT}}$:$m_{\text{polyvinylpyrrolidone}}$ = 1:3) by stirring for 30 min and ultrasonication for 10 min in an ice bath using an Ultrasonic Cell Disruption System with 60% power (1500 W). Then, the solution was treated by freeze dryer to obtain sa-SWNT powder. Subsequently, the optimized amount of sa-SWNT powder was dispersed in DMAc using an Ultrasonic Cell Disruption System with 600 W for 15 min in an ice bath under $N_2$ flow (the weight percentage of sa-SWNTs is calculated as the ratio of the mass of sa-SWNTs to the mass of heterocyclic aramid after polymerization). Then, the obtained homogenous sa-SWNT/DMAc dispersion was slowly added to the mixture containing LiCl (3.5 wt%) and DMAc. Subsequently, the optimized amounts of PPD and PABZ were added to the reaction system, respectively, and the mixture was kept stirring for about 1 h. After cooling the mixture to below 10 °C, the TPC was added, and then the reaction was conducted under stirring for 1.5 h. After the reaction, a black viscous spinning dope with a dynamic viscosity of 40,000–60,000 mPa·s was obtained. As a control sample, the pure heterocyclic aramid spinning dope was obtained following a similar preparation procedure without adding the sa-SWNT powder. The al-SWNT/heterocyclic aramid, as-SWNT/heterocyclic aramid and sc-SWNT/heterocyclic aramid spinning dopes were prepared under the same procedures by adding al-SWNT, as-SWNT and sc-SWNT powder, respectively.

## Fabrication of sa-SWNT-HAFs through wet-spinning

The obtained sa-SWNT/heterocyclic aramid spinning dope was poured into a degassing tank equipped with a metering pump and a nitrogen inlet. After a vacuum-defoaming treatment, the spinning dope was transported to a spinning pot on spinning line. After extrusion by a spinning pump under high-pressure nitrogen, the spinning dope was transported to spinneret plate, and injected into multistage coagulation baths with different concentrations (primary coagulation bath, $m_{\text{water}}$:$m_{\text{DMAc}}$ = 1:1; secondary coagulation bath, $m_{\text{water}}$:$m_{\text{DMAc}}$ = 4:1). The total drawing ratio in two coagulation process is 2. The primary fibers were obtained after the washing with flowing deionized water and drying processes (120 °C). Subsequently, the sa-SWNT-HAFs were obtained after a heating treatment in the $N_2$ atmosphere (410 °C) and collected via an automatic winding device. The HAFs, al-SWNT-HAFs, as-SWNT-HAFs, and sc-SWNT-HAFs were prepared under the same procedures by adding corresponding spinning dope.

## Preparation of sa-SWNT-HAF nanofibers

The sa-SWNT-HAFs (0.04 g) and KOH (0.06 g) were added to DMSO (20 mL)[52]. Then the mixture was magnetically stirred for fixed times. The sc-SWNT-HAF nanofibers and HAF nanofibers were prepared under the same procedures.

## Strength measurement method at high loading rates

The strengths of the fibers at different loading strain rates were measured by a mini split Hopkinson tensile bar (mini-SHTB, Supplementary Fig. 28)[53]. While a sleeve-typed bullet launched by a gas gun strikes the mass block fixed at the end of the incident bar, a tensile wave is generated and propagates along the incident bar. When it reaches the specimen clamped at the end of the incident bar, the dynamic tension is applied on the single fiber specimen, causing its ultimate failure.

A high-frequency response quartz piezoelectric force sensor (Kistler 9001, 180 kHz) is used to directly measure the tensile force of the single fiber. Due to the sufficient high impact impedance of the incident bar, the end of the incident bar could be considered as a free interface when compared to the single fiber. Therefore, the strain rate, the strain, and the strength of the fiber specimen can be determined[53],

$$\dot{\varepsilon}(t) = 2C_0 \frac{\varepsilon_{\text{I}}}{l_{\text{S}}} \tag{2}$$

$$\varepsilon(t) = 2C_0 \int_0^t \frac{\varepsilon_{\text{I}}}{l_{\text{S}}} \mathrm{d}t \tag{3}$$

$$\sigma(t) = \frac{F}{A_{\text{S}}} \tag{4}$$

where $C_0$ is the elastic wave velocity, $l_{\text{S}}$ and $A_{\text{S}}$ are the length and the cross-sectional area of the specimen, respectively, $\varepsilon_{\text{I}}$ is the measured incident strain, and $F$ is the force collected by the force sensor.

The typical incident wave and force signals of a single fiber are shown in Supplementary Fig. 29a, and the corresponding engineering stress versus strain curves are shown in Supplementary Fig. 29b. The strain rate attains the constant value when the strain exceeds 0.01, validating the experimental method for measuring the dynamic strength of single fibers. The typical engineering stress versus strain curves of HAFs, sc-SWNT-HAFs, and sa-SWNT-HAFs under various strain rates are shown in Supplementary Fig. 29c, showing the almost brittle mechanical behavior of these fibers.

## Laser-induced impact testing

The schematic diagram of the laser-induced impact testing on fibers is shown in Supplementary Fig. 32. The laser ablates the 100 nm thick gold film to create a fast expanding plasma on the surface, causing the fast expansion of the 76 µm thick polydimethylsiloxane (PDMS) layer, which impacts the single fiber specimen with high velocity. To obtain a clear process of the impact process, the $5 \times 10^6$ fps high-speed video camera (KIRANA UHS Camera) and the SI-LUX640 automatic laser lighting system are used in experiments. One end of the single fiber specimen is fixed, and the other end suspends a small mass (0.22 g) to ensure its straightness. The pre-tension force is estimated to be 2.2 mN, which is negligible in the experiments. Supplementary Fig. 33 shows the typical impact process of several fibers. The impact velocities and the transverse wave velocities of the single fibers are calculated using the first two frames and the last two frames, respectively.

## Molecular dynamics simulations

The microstructural evolution and mechanical behaviors of materials were explored by molecular dynamics (MD) simulations using large-scale atomic/molecular massively parallel simulator (LAMMPS) computational package[54]. The polymer consistent force field (PCFF) was adopted to describe the interatomic potentials[55,56]. The long-range Columbia interaction was included using particle-particle-particle mesh (PPPM) method[57], while the van der Waals interaction was described by the 9-6 Lennard-Jones potential. To integrate the Newton equations of motion, the Verlet algorithm was adopted with a time step of 1 fs. Before the tensile deformation protocol was started, the structures were energy minimized using a conjugate gradient algorithm. In tensile simulations, the model of HAFs was formed by arranging heterocyclic aramid chains in a brick-wall manner, while the models of sc-SWNT-HAFs and sa-SWNT-HAFs were formed by additionally inserting sc-SWNTs and sa-SWNTs, respectively. Periodic boundary conditions were used along all directions. To ensure quasi-static loading, the uniaxial tensile strain was applied by uniaxially deforming the periodic simulation box at an engineering strain rate of $1 \times 10^8$ s$^{-1}$.

To investigate the influence range of the $sp^2$ carbon sheet on the heterocyclic aramid chains, 50 layers of heterocyclic aramid chains were placed onto a planar $sp^2$ carbon sheet for reducing computational costs. Periodic boundary conditions were used along the in-plane directions, and the vacuum layer with a thickness over 10 nm was adopted along the out-of-plane direction. During the thermostat process, the $sp^2$ carbon sheet was fixed. The temperature increased from nearly 0 K to 1000 K and then decreased to room temperature of 298 K, which was then equilibrated for 2 ns (Supplementary Fig. 35c, d).

To calculate the binding energy between typical polymer chains and the $sp^2$ carbon sheet, heterocyclic aramid, $C_2H_2$, polystyrene (PS), polyethylene (PE), polyvinyl alcohol (PVA), benzene chains, and polyvinylpyrrolidone were placed onto an $sp^2$ carbon sheet (Supplementary Note 3, Supplementary Fig. 35b). Periodic boundary conditions were used along the in-plane directions, and the vacuum layer with a thickness over 2 nm was adopted along the out-of-plane direction. The binding energy is the sum of the energy of the isolated polymer chain and the energy of the isolated $sp^2$ carbon sheet minus the total energy of the composite system (normalized per atom).

## DFTB calculations

To calculate the Young's modulus of crystalline heterocyclic aramid (Supplementary Fig. 35a), density functional theory-based tight-binding (DFTB) calculations were performed using DFTB+ package[58]. Periodic boundary condition was adopted along all directions. All DFTB computations were performed with DFT-D3 dispersion correction with Becke-Johnson damping[59,60]. The 3-ob-1 Slater-Koster set of parameters was employed[61–63]. Energy-minimized tensile tests on crystalline heterocyclic aramid were performed using a conjugate gradient algorithm with a force-threshold criterion of $10^{-4}$ Hartree/Bohr.

## Theoretical analyses

The HAFs were divided into crystalline and amorphous regions, and the weight fraction of amorphous region ($f_a$) can be calculated as

$$f_a = \frac{E_{\rho,c} - E_\rho}{E_{\rho,c} - E_{\rho,a}} \tag{5}$$

where $E_\rho$, $E_{\rho,a}$, and $E_{\rho,c}$ represent the gravimetric moduli of HAF, amorphous heterocyclic aramid, and crystalline heterocyclic aramid, respectively. With the gravimetric moduli of crystalline heterocyclic aramid (about 168 GPa g$^{-1}$ cm$^3$) from DFTB calculations, amorphous heterocyclic aramid (about 36 GPa g$^{-1}$ cm$^3$) and HAF in experiments (about 86 GPa g$^{-1}$ cm$^3$), $f_a$ can be estimated as 61.9%. Further, we propose a modified rule-of-mixture to predict the theoretical gravimetric modulus of sa-SWNT-HAFs as

$$E_{\rho,ssh} = E_{\rho,CNT} \times f_{CNT} + E_{\rho,c} \times (1 - f_a + f) + E_{\rho,a} \times (f_a - f_{CNT} - f) \tag{6}$$

where $E_{\rho,ssh}$ and $E_{\rho,CNT}$ are the gravimetric moduli of sa-SWNT-HAF and SWNT, $f_{CNT}$ and $f$ represent the weight fraction of SWNTs and the increased weight fraction of crystalline region induced by SWNTs. Considering that SWNTs can act as nucleating agents for polymer crystallization and templates for polymer orientation, the influence range of amorphous heterocyclic aramid into crystalline heterocyclic aramid induced by SWNT is expected to be far larger than the size of SWNT. Based on the enhancement (13%) in modulus from HAFs to sa-SWNT-HAFs, $f$ can be calculated as 8.6%, which corresponds to an influence range of 12.2 nm.

## Data availability

The data sets within the paper and Supplementary Information of the current study are available from the authors upon request. Source data are provided with this paper.

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

## Acknowledgements

We thank X. Sha, R. Yang, and L. Meng at Zeiss, for their help of FIB-SEM measurements. We thank Y. Shao, S. Wang, S. Xu and J. Li for helpful discussions. We Thank the beamline BL19U2 of National Centre for Protein Science Shanghai at Shanghai Synchrotron Radiation Facility for conducting the SAXS characterizations. We acknowledge support from the Beijing Municipal Science and Technology Commission (2018YFA0703502), the National Natural Science Foundation of China

(52021006, 51720105003, 21790052, 52102035, 12272391, 12232020), the Strategic Priority Research Program of CAS (XDB36030100), and the Beijing National Laboratory for Molecular Sciences (BNLMS-CXTD-202001).

## Author contributions

J.L., Y.W., X.J., and X.L. contributed equally to this work. JinZ., K.J., E.G., and X.W. conceived and supervised the project. J.L. and Y.W. performed the preparation and characterization of fibers, with assistance from Z.G., M.J., Z.X., L.L., and T.L.. X.J. performed the theoretical calculations and simulations, with assistance from E.G. X.L. performed dynamic mechanical tests, with assistance from X.W., J.L., Jiangw.Z. and H.D. performed the test of SAXS. All authors contributed to discussions and commented on the paper.

## Competing interests

The authors declare no competing interests.
