## [Peer Review File · Nature Communications]

REVIEWER COMMENTS

Reviewer #1 (Remarks to the Author):

J. Luo et al. reported mechanical strong aramid fibers by small addition of CNTs. The manuscript seems to be interesting and the properties were significantly improved compared to conventional aramid fibers. The conclusion was well supported by data. I would recommend this article for publication in Nature Communications with minor revision.

1.They should provide basic information about SWCNT in terms of length(aspect ratio) and crystallinity(IG/ID). They just mentioned purity (>95%) and did not mentioned supplier.

2.In fig1, sa-SWCNT shows end-functionalized SWCNT. What is the ratio of end and side-functionalization?

3.Why does sa-SWCNT-HAF show higher degree of orientation compared to sc-SWCNT, resulting in better properties?

4.In supporting information Table12, there are many references including graphene fibers, and CNT fibers. However, there are some mistakes reporting mechanical properties and density of fiber in case of ref 25. The tensile modulus of thermally annealed CNT fibers at 1700 C deg should be 368 GPa, rather than 3.68 GPa. In addition, I would recommend they need to cite several recent references of mechanically strong CNT/GO and CNT-polymer based composite carbon fibers.

Reviewer #2 (Remarks to the Author):

This is an interesting paper upon the preparation of strong and tough aramid fibres through the incorporation of a small amount of carbon nanotubes. It should be published after some minor modifications.

Title: It would be better to state "strong and tough" rather than "strongest and toughest". People might dispute this claim.

Abstract: The values of strength and toughness quoted are confusing. I realise that they are the highest values obtained but the way it is written implies that they are the increase in values. It would also be interesting to quote the increase in modulus values in the abstract.

p.5: It is claimed that the SWNTs break but the length of the broken tubes is not given as far as I can tell. Is it possible to give values of tube length?

p.11: The analysis of deformation mechanisms is weak. It is certainly true that the modulus values obtained are above those predicted by the rule of mixtures. There is one study upon a similar system that ought to be cited (Deng - doi:10.1016/j.polymer.2010.02.040). They used Raman spectroscopy to follow stress transfer from nanotubes to an aramid fiber. The authors should try to measure Raman band shifts in their fibers or at least suggest it as a method to understand the deformation mechanisms. This would be much more useful than their computational analysis.

Point-by-point response to the reviewers' comments

Reviewer #1 (Remarks to the Author):

J. Luo et al. reported mechanical strong aramid fibers by small addition of CNTs. The manuscript seems to be interesting and the properties were significantly improved compared to conventional aramid fibers. The conclusion was well supported by data. I would recommend this article for publication in Nature Communications with minor revision.

Reply:

Reply: Thank you very much for your recommendation and valuable comments. We improved our work accordingly. The revisions were shown with highlights in the revised manuscript.

Comment 1: They should provide basic information about SWCNT in terms of length(aspect ratio) and crystallinity(I_G/I_D). They just mentioned purity (>95%) and did not mentioned supplier.

Reply:

Thank you for this valuable comment. In response, we provided this detailed information in the main text and supplementary information. The raw SWNTs were purchased from Jiangsu XFNANO Materials Tech Co., Ltd. According to your advice, we conducted a statistical analysis of the length of the raw SWNTs. The results show that the average length of raw SWNTs is around 6.72 μm (**Supplementary Fig. 2**). The crystallinity (I_G/I_D) of different SWNTs is shown in **Supplementary Fig. 5**. The average I_G/I_D value of raw SWNTs is 43.8.

Revision:

In the main text:

Line 3-17 in Page 5, we added the description of the length of SWNTs.

“The transmission electron microscope (TEM) and scanning electron microscope (SEM) images of raw SWNTs indicate that they have a diameter of around 3 nm and an average length of around 6.72 μm (**Fig. 2b, c, Supplementary Figs. 1a, 2a**). These raw SWNTs prefer to be bundled because of their large aspect ratio and strong intertube interaction. Through the process of primary oxidation, long-SWNTs (average length of 1.66 μm) decorated with oxygen-containing functional groups were obtained (**Supplementary Figs. 1b, e, 2b, 3a**). After violent ultrasonication of the long-SWNT dispersion, the length of SWNTs was further reduced, and the resultant short-SWNTs (average length of 0.66 μm) exhibit a good dispersity (**Supplementary Figs. 1c, 2c, 4**). The reoxidation of short-SWNTs was performed to prepare short carboxyl SWNTs (sc-SWNTs, average length of 0.63 μm) with more reactive carboxyl groups (**Supplementary Figs. 1d, 2d, 3, 4**). In order to construct strong covalent bonding between SWNTs and heterocyclic aramid chains, amino groups which can react with heterocyclic aramid monomers were introduced to SWNTs. Thus, sa-SWNTs were prepared by carrying out an amination reaction of sc-SWNTs with ethylenediamine (**Supplementary Fig. 3e**). The sa-SWNTs show a good dispersity and their lengths most range in 0.4–1.0 μm (average length of 0.62 μm), which are near the persistence length (**Fig. 2d, Supplementary Figs. 1, 2f, 4 and Supplementary Note 1**)³⁶.”

In the supplementary information:

Line 7 in Page 2, the supplier of SWNTs was provided.

“The raw SWNTs were purchased from Jiangsu XFNANO Materials Tech Co., Ltd.”

Page 13 and Page 16, the supplementary figures showing the average length and the crystallinity (I_G/I_D) of different SWNTs were provided.

Supplementary Fig. 2 | Length distribution of SWNTs. The length distribution of (a) raw SWNTs, (b) long-SWNTs, (c) short-SWNTs, (d) sc-SWNTs, and (e) sa-SWNTs.

Supplementary Fig. 5 | Comparison of the I_G/I_D values of different SWNTs from Raman spectra analysis.

Comment 2: In fig1, sa-SWCNT shows end-functionalized SWCNT. What is the ratio of end and side-functionalization?

Reply:

Thank you for this insightful comment. Due to the similar chemical properties of carbon atoms at the ends and carbon atoms in the walls, it is difficult to differentiate the end and side-functional groups of SWNT by current characterization techniques, such as infrared spectrometer, NMR spectroscopy, X-ray photoelectron spectroscopy. In our work, we used a high-power cell disruptor to ultrasonically break SWNTs for a long time. In this process, SWNTs break at the defect as much as possible so that they are subsequently functionalized at the ends. However, it is impossible to break SWNTs from every defect into segments. Therefore, some defects remain on the wall of SWNTs, leading to the presence of side-functionalization. However, the ratio of the intensities of the G and D peaks calculated from Raman spectrum of treated SWNTs are still as high as about 15, which indicates that their tube walls remain relatively intact (**Supplementary Fig. 5**). Thus, we speculate that most functional groups are at the ends while few on the sides of walls.

Comment 3: Why does sa-SWCNT-HAF show higher degree of orientation compared to sc-SWCNT, resulting in better properties?

Reply:

Thank you for this insightful comment. Compared to sc-SWNT-HAFs, the amino groups of sa-SWNTs can form covalent bonds with heterocyclic aramid chains. During the spinning process, the covalent bonding between heterocyclic aramid chains and sa-SWNTs is favorable to improve the alignment of sa-SWNTs because of effective stretching. The alignment of sa-SWNTs can further induce the alignment of heterocyclic aramid chains around sa-SWNTs. Consequently, sa-SWNT-HAFs have higher degree of crystallinity and orientation, and more ordered arrangement of microfibers than sc-SWNT-HAFs (**Fig. 3f-i, j, k**). On the other hand, the covalent bonding between heterocyclic aramid chains and sa-SWNTs suppresses the crack nucleation/growth. Hence, sa-SWNT-HAFs exhibit better properties.

Comment 4: In supporting information Table12, there are many references including graphene fibers, and CNT fibers. However, there are some mistakes reporting mechanical properties and density of fiber in case of ref 25. The tensile modulus of thermally annealed CNT fibers at 1700 C deg should be 368 GPa, rather than 3.68 GPa. In addition, I would recommend they need to cite several recent references of mechanically strong CNT/GO and CNT-polymer based composite carbon fibers.

Reply:

Thanks for your valuable suggestion. According to your advice, we scrutinized these data and made corrections for these mistakes. In addition, we cited some recent references of mechanically strong CNT/GO hybrid fibers and CNT-polymer based composite carbon fibers.

Revision:

In the supplementary information:

Page 58-Page 60, the supplementary table showing the data for monofilaments of our prepared fibers and fibers reported in literature was revised.

Supplementary Table 12 | Comparisons of mechanical properties of our fibers, carbon fibers, graphene fibers, CNT fibers, and polymer fibers (mechanical properties were collected for monofilaments).

Materials	Strength (GPa)	Modulus (GPa)	Elongation at break (%)	Toughness (MJ m ⁻³)	Density (g cm ⁻³)	Gauge length (mm)	Loading rate (mm min ⁻¹)	Ref.
Kevlar	4	120	3	~70	1.50	--	15	9
Kevlar-CNT	5	130	3.5	~90	1.50	--	15	9
PBO	2.6	138	2	~26	--	25.4	5	10
PBO-CNT	4.2	167	2.8	~58	--	25.4	5	10
UHMWPE	3.51	122.6	4.03	~75	0.97	--	2.54	13
UHMWPE-CNT	4.17	136.8	4.65	~105	0.995	--	2.54	13
Kevlar 29	2.47	84.5	3.2	~45	1.45	30	4.07	14
Heterocyclic aramid fiber ¹	6.1	150.1	4.35	~135	--	20	10	4
Heterocyclic aramid fiber ²	5.81	143.2	4.15	~125	--	20	1	15
Graphene fiber ¹	0.22	--	39	~46.3	0.8	7	0.06	16
Graphene fiber ²	0.501	11.2	6.7	~18	--	--	10% min ⁻¹	17
Graphene fiber ³	1.78	385	0.5	~5	--	5	10% min ⁻¹	18
Graphene fiber ⁴	1.9	309	0.67	~6.5	--	20	0.5	19

Graphene fiber ⁵	3.4	341.7	1	~20	1.9	5	--	20
Graphene fiber ⁶	1.08	77.6	1.45	~10	1.74	--	0.5	21
CNT fiber ¹	4.2	260	3.5	~80	--	--	--	22
CNT fiber ²	1.9	195	5.28	~61.8	0.2	10	3	23
CNT fiber ³	4.04	83.3	6.01	~125	--	10	0.6	24
CNT fiber ⁴	6.57	629	1.53	~55	1.71	25	--	25
CNT-PI fiber	4.8	390	4.1	~128	1.78	25	2	26
	6.21	528	1.69	~85	1.74	25	2	26
Carbon fiber	1.24	13.5	28	~161.7	1.18	10	5	27
Graphene/Carbon fiber ¹	2.44	358.3	0.7	~15	1.9	5	--	28
Graphene/Carbon fiber ²	1.92	233	1.1	~11	~1.6	20	10 ⁻⁴	29
Graphene oxide/Carbon fiber ¹	1.1	100	1.27	6.96	1.43-1.69	25	2.5	30
Graphene oxide/Carbon fiber ²	2.12	138	1.53	~20	1.73	25	2	31
CNT/ Carbon fiber ²	2.2	60	4	~50	1.48	10	1	32
CNT/Graphene oxide fiber	6.05	422	3.6	154	2.01	25	2	33

CNT/Graphene oxide fiber	0.69	--	0.6	~3	1.6	25	2.5	34
HAF (This work)	5.12	125.1	4.27	110.7	1.44	20	1	--
sa-SWNT-HAF (This work)	6.44	141.7	5.37	184.0	1.45	20	1	--

Line 14-16 in Page 65 and Line 21-31 in Page 65, we added relevant literature.

“26. Kim, S. G. et al., Ultrahigh strength and modulus of polyimide-carbon nanotube based carbon and graphitic fibers with superior electrical and thermal conductivities for advanced composite applications. *Compos. Pt. B-Eng.* **247**, 110342 (2022).

29. Gao, Z. et al., Graphene reinforced carbon fibers. *Sci. Adv.* **6**, eaaz4191 (2020)

30. Eom, W. et al., Microstructure-controlled polyacrylonitrile/graphene fibers over 1 gigapascal strength. *ACS Nano* **15**, 13055-13064 (2021).

31. Kim, J. et al., Longitudinal alignment effect of graphene oxide nanoribbon on properties of polyimide-based carbon fibers. *Carbon* **198**, 219-229 (2022).

32. Li, M. et al., Robust carbon nanotube composite fibers: strong resistivities to protonation, oxidation, and ultrasonication. *Carbon* **146**, 627-635 (2019).

33. Kim, S. G. et al., Ultrastrong hybrid fibers with tunable macromolecular interfaces of graphene oxide and carbon nanotube for multifunctional applications. *Adv. Sci.* **9**, 2203008 (2022).

34. Eom, W. et al., Carbon nanotube-reduced graphene oxide fiber with high torsional strength from rheological hierarchy control. *Nat. Commun.* **12**, 396 (2021).”

Reviewer #2 (Remarks to the Author):

This is an interesting paper upon the preparation of strong and tough aramid fibres through the incorporation of a small amount of carbon nanotubes. It should be published after some minor modifications.

Reply: Thank you very much for your valuable comments and suggestions. We improved our work accordingly. The revisions were shown with highlights in the revised manuscript.

Comment 1: Title: It would be better to state "strong and tough" rather than "strongest and toughest". People might dispute this claim.

Reply:

Thank you for this comment. In response, we revised the title to “Fabricating Strong and Tough Aramid Fibers by Small Addition of Carbon Nanotubes”.

Revision:

In the main text:

Line 1-2 in Page 1, we revised this statement of “strongest and toughest” to “strong and tough”. “Fabricating Strong and Tough Aramid Fibers by Small Addition of Carbon Nanotubes”

Comment 2: Abstract: The values of strength and toughness quoted are confusing. I realise that they are the highest values obtained but the way it is written implies that they are the increase in values. It would also be interesting to quote the increase in modulus values in the abstract.

Reply:

Thank you for this useful suggestion. In response, we revised the description of “Herein, we report a simultaneous improvement in strength and toughness of heterocyclic aramid fibers by 26% (6.44 ± 0.11 GPa) and 66% (184.0 ± 11.4 MJ m⁻³), respectively, via *in situ* polymerizing small amount (0.05 wt%) of short aminated single-walled carbon nanotubes (SWNTs) into heterocyclic aramid fibers.” to “Herein, we report a simultaneous improvement in strength, toughness, and modulus of heterocyclic aramid fibers by 26%, 66%, and 13%, respectively, via *in situ* polymerizing small amount (0.05 wt%) of short aminated single-walled carbon nanotubes (SWNTs) into heterocyclic aramid fibers, achieving a tensile strength of 6.44 ± 0.11 GPa, a toughness of 184.0 ± 11.4 MJ m⁻³, and a Young’s modulus of 141.7 ± 4.0 GPa.”

Revision:

In the main text:

Line 23-27 in Page 1, we revised the description and added the increase in modulus.

“Herein, we report a simultaneous improvement in strength, toughness, and modulus of heterocyclic aramid fibers by 26%, 66%, and 13%, respectively, via *in situ* polymerizing small amount (0.05 wt%) of short aminated single-walled carbon nanotubes (SWNTs) into heterocyclic aramid fibers, achieving a tensile strength of 6.44 ± 0.11 GPa, a toughness of 184.0 ± 11.4 MJ m⁻³, and a Young’s modulus of 141.7 ± 4.0 GPa.”

Comment 3: p.5: It is claimed that the SWNTs break but the length of the broken tubes is not given as far as I can tell. Is it possible to give values of tube length?

Reply:

Thank you for this valuable comment. In response, we conducted a statistical analysis of the length of SWNTs. The results show that the average length of raw SWNTs, long-SWNTs, short-SWNTs, sc-SWNTs, and sa-SWNTs is around 6.72 μm , 1.66 μm , 0.66 μm , 0.63 μm , and 0.62 μm , respectively (**Supplementary Fig. 2**). we provided this detailed information in the main text and supplementary information.

Revision:

In the main text:

Line 3-17 in Page 5, we added the description of the length of SWNTs.

“The transmission electron microscope (TEM) and scanning electron microscope (SEM) images of raw SWNTs indicate that they have a diameter of around 3 nm and an average length of around 6.72 μm (**Fig. 2b, c, Supplementary Figs. 1a, 2a**). These raw SWNTs prefer to be bundled because of their large aspect ratio and strong intertube interaction. Through the process of primary oxidation, long-SWNTs (average length of 1.66 μm) decorated with oxygen-containing functional groups were obtained (**Supplementary Figs. 1b, e, 2b, 3a**). After violent ultrasonication of the long-SWNT dispersion, the length of SWNTs was further reduced, and the resultant short-SWNTs (average length of 0.66 μm) exhibit a good dispersity (**Supplementary Figs. 1c, 2c, 4**). The reoxidation of short-SWNTs was performed to prepare short carboxyl SWNTs (sc-SWNTs, average length of 0.63 μm) with more reactive carboxyl groups (**Supplementary Figs. 1d, 2d, 3, 4**). In order to construct strong covalent bonding between SWNTs and heterocyclic aramid chains, amino groups which can react with heterocyclic aramid monomers were introduced to SWNTs. Thus, sa-SWNTs were prepared by carrying out an amination reaction of sc-SWNTs with ethylenediamine (**Supplementary Fig. 3e**). The sa-SWNTs show a good dispersity and their lengths most range in 0.4–1.0 μm (average length of 0.62 μm), which are near the persistence length (**Fig. 2d, Supplementary Figs. 1, 2f, 4 and Supplementary Note 1**)³⁶.”

In the supplementary information:

Page 13, the supplementary figure showing the average length of different SWNTs was provided.

Supplementary Fig. 2 | Length distribution of SWNTs. The length distribution of (a) raw SWNTs, (b) long-SWNTs, (c) short-SWNTs, (d) sc-SWNTs, and (e) sa-SWNTs.

Comment 4: p.11: The analysis of deformation mechanisms is weak. It is certainly true that the modulus values obtained are above those predicted by the rule of mixtures. There is one study upon a similar system that ought to be cited (Deng - doi:10.1016/j.polymer.2010.02.040). They used Raman spectroscopy to follow stress transfer from nanotubes to an aramid fiber. The authors should try to measure Raman band shifts in their fibers or at least suggest it as a method to understand the deformation mechanisms. This would be much more useful than their computational analysis.

Reply:

Thank you for this useful suggestion. In response, we cited this literature and added a description of Raman spectroscopy. We fully agreed that Raman spectroscopy is a powerful tool to follow the stress transfer between nanotubes and polymers, and we tried to measure Raman band shifts during *in situ* tests of the fibers. However, due to the low content of SWNTs (0.05 wt%) and high fluorescence intensity of HAF in our work, it is difficult to detect the Raman signals of SWNTs in our composite fibers. In this literature, PPTA has strong Raman signals. In addition, PPTA/SWNT with 0.5% content of SWNT also exhibits obvious G'-band of SWNT. Therefore, Raman spectroscopy can be used to study stress transfer from SWNTs to PPTA by observing the variation of band shift of SWNT G'-band in this literature. Specifically, heterocyclic aramid fiber (HAF) in our work was formed by the copolymerization of three monomers (p-phenylenediamine, terephthaloyl chloride and 5-(6)-Amino-2-(4-aminobenzene)benzimidazole). Due to the addition of **heterocyclic monomer** (5-(6)-Amino-2-(4-aminobenzene)benzimidazole), the fluorescence signal of HAF is much stronger than that of PPTA fiber, which seriously affects the detection of Raman signals. We performed Raman characterization of our fibers using **532 nm** laser and **633 nm** laser, respectively. No significant Raman signal was observed for HAF and sa-SWNT-HAF compared with PPTA fiber due to the interference of fluorescence signals (**Fig. R1a-d**). Subsequently, to eliminate the effect of fluorescence of HAF as much as possible, we chose **1064 nm** laser as the excitation wavelength of Raman. The test results showed that the Raman signals of HAF, sa-SWNT-HAF, and PPTA fibers were all enhanced (**Fig. R1e, f**). However, we still could not find G'-band of SWNT in the sa-SWNT-HAF, which is crucial for studying stress transfer from SWNTs to HAF. We speculated that there are two reasons why G'-band of SWNT could not be detected: (1) the low content of SWNTs. The content of SWNTs in the sa-SWNT-HAF is only **0.05 wt%**, which is much lower than that reported in this literature (**0.5 wt%**). (2) the low energy of 1064 nm laser. The energy of 1064 nm laser is weaker than that of 532 nm and 633 nm lasers, making it difficult to detect signals of SWNTs. It should be noted that although we cannot follow the stress transfer from SWNTs to HAF by Raman spectroscopy, we provided other experimental evidence and atomistic simulation to prove the strong interfacial interactions in sa-SWNT-HAFs to achieve efficient stress transfer, including its high resistance to the interchain slippage, high resistance to dissolution into nanofibers (**Fig. 3I, Supplementary Fig. 19**), and higher binding energy between sa-SWNTs and heterocyclic aramid chains than that between sa-SWNTs and other polymers (**Fig. 5c**).

Revision:

In the main text:

Line 11-13 in Page 12, we added a description about Raman spectroscopy.

“Raman spectroscopy is commonly used to understand the enhancement mechanism of carbon-based composites^{12,48}. However, due to the low content of SWNTs (0.05 wt%) and high fluorescence intensity of HAF in our work, it is difficult to detect the Raman signals of SWNTs in our composite fibers.”

Line 19-20 in Page 21, we added this literature.

“48. Deng, L., Young, R. J., van der Zwaag, S., & Picken, S. Characterization of the adhesion of single-walled carbon nanotubes in poly (p-phenylene terephthalamide) composite fibres. *Polymer* **51**, 2033-2039 (2010).”

Fig. R1 | Raman spectra and fluorescence spectrum of different fibers. Raman spectra with (a) 532 nm and (b) 633 nm laser of HAF and sa-SWNT-HAF. c, Raman spectra with different lasers of PPTA fiber. d, Fluorescence spectrum of HAF. e, Raman spectra with 1064 nm laser of HAF and sa-SWNT-HAF. f, Raman spectrum with 1064 nm laser of PPTA fiber.

REVIEWERS' COMMENTS

Reviewer #1 (Remarks to the Author):

I would recommend this revised article for publication in Nature Comm. All response was supported by reasonable data and comments.

Reviewer #2 (Remarks to the Author):

The changes that the authors have made in response to the comments of both Referees now make the paper suitable for publication.